# Referential equations for pulmonary diffusing capacity using GAMLSS models derived from Japanese individuals with near-normal lung function

Yosuke Wada[1]*, Norihiko Goto[1], Yoshiaki Kitaguchi[1], Masanori Yasuo[2], Masayuki Hanaoka[1]

**1** First Department of Internal Medicine, Shinshu University School of Medicine, Matsumoto, Nagano, Japan, **2** Departments of Clinical Laboratory Sciences, Shinshu University School of Health Sciences, Matsumoto, Nagano, Japan

* yosuke@shinshu-u.ac.jp

**Data Availability Statement:** All relevant data are within the paper and its Supporting Information files.

## Abstract

### Objective

To generate appropriate reference values for the single-breath diffusing capacity of the lungs for carbon monoxide ($D_{LCO}$), alveolar volume ($V_A$), and the transfer coefficient of the lungs for carbon monoxide ($K_{CO}$, often denoted as $D_{LCO}/V_A$) in the Japanese population. We also intended to assess the applicability of these values for the Japanese population by comparing them to those published by the Global Lung Function Initiative in 2017 (GLI-2017) and previous values.

### Methods

In this retrospective study, we measured the spirometric indices, $D_{LCO}$, $V_A$, and $K_{CO}$ of the Japanese population aged 16–85 years. The lambda, mu, and sigma (LMS) method and the generalized additive models for the location, scale, and shape program in R were used to generate the reference values.

### Results

We conducted a total of 390 tests. The GLI-2017 z-scores of $D_{LCO}$ were approximately zero, whereas those of $K_{CO}$ and $V_A$ were far from zero. In the present study, the mean square errors of the $D_{LCO}$, $V_A$, and $K_{CO}$ reference values were lower than the reference values derived from GLI-2017 and previous linear regression equations.

### Conclusions

Reference values obtained in this study were more appropriate for our sample than those reported in GLI-2017. Differences between the two equations were attributed to underestimating $K_{CO}$ ($D_{LCO} / V_A$) and overestimating $V_A$, respectively, by the GLI-2017 for the Japanese population.

**Funding:** The authors received no specific funding for this work.

**Competing interests:** The authors have declared that no competing interests exist.

## Introduction

Single-breath diffusing capacity of the lungs for carbon monoxide ($D_{LCO}$) is a simple non-invasive method for diagnosing and monitoring patients with chronic lung diseases, such as chronic obstructive pulmonary disease (COPD) or interstitial lung disease (ILD) [1]. $D_{LCO}$ is a commonly used indicator for the early detection and monitoring of chronic lung diseases. However, there are no standardized reference values for the $D_{LCO}$, alveolar volume ($V_A$), and the transfer coefficient of the lung for carbon monoxide ($K_{CO}$, often denoted as $D_{LCO}/V_A$) in the Japanese population.

In 2017, the Global Lung Function Initiative (GLI) published new $D_{LCO}$ reference values for Caucasians aged 5–85 years (GLI-2017) [2]. Three retrospective studies have assessed the GLI-2017 reference values in various population sets of both healthy controls and patients [3–5]. The GLI-2017 reference values were based on the lambda, mu, and sigma (LMS) method, which used the generalized additive models of the location, shape, and scale (GAMLSS) package in the statistical program R [6]. In 2020, the GLI updated reference values for lung function tests (LFTs) in individuals of European ancestry using the LMS method of GAMLSS [7]. The GAMLSS modeling approach is suitable for deriving reference values for lung function outcomes [6, 8, 9]. Despite no standardized reference values for the $D_{LCO}$ in the Japanese population derived from GAMLSS modeling, researchers have reported on values for LFTs using GAMLSS modeling [10].

The European Respiratory Society (ERS) and the American Thoracic Society (ATS) have updated their standards for measuring carbon monoxide gas transfer in the lungs, and additional guidelines for the technique are available [11, 12]. However, there is no agreement on the best equations for various ethnic groups.

We aimed to develop GAMLSS models using collated contemporary $D_{LCO}$ ($T_{LCO}$) data from Japanese patients without chronic lung diseases, such as COPD or ILD, which reduces diffusivity, and derive the reference values for $D_{LCO}$ measurements. Next, we intended to examine if our predicted values differed less from those obtained from the frequently used GLI-2017 or linear prediction equations.

## Materials and methods

### Study participants

This retrospective observational study was approved by the Ethics Committee of the Shinshu University (permission number 5139) and was performed in accordance with the principles outlined in the tenets of the Declaration of Helsinki of the World Medical Association. The requirement for written informed consent was waived owing to the use of de-identified retrospective data. Contrarily, this research used an opt-out consent model, which allowed the participants to withdraw their consent at any time and have their information deleted from the registry.

The inclusion criteria were as follows: (1) Japanese patients without chronic lung diseases, such as COPD or ILD, and derived reference values for $D_{LCO}$ ($T_{LCO}$) measurements at the first medical consultation in our institute (Shinshu University Hospital, Matsumoto, Japan) from January 2008 to November 2021, (2) never smoker, (3) age 16–85 years, (4) body mass index (BMI) $<30$ kg/m$^2$, (5) no abnormality or localized shadow based on chest computed tomography (CT) performed within 6 months before the LFT, (6) the percent $D_{LCO}$ $>80\%$ according to the prediction equations of Nishida et al. and Burrows et al., and (7) ambulant patients [13, 14]. To ensure a sufficient sample size, patients with early-stage lung cancer, sarcoidosis, or asthma, with small abnormal shadows that did not meet the exclusion criteria or without

abnormal shadows were included in the CT screening. It was difficult to include all participants with abnormalities in the CT screening considering the study's retrospective design. The exclusion criteria were as follows: (1) cardiovascular disease other than hypertension, (2) motor neuron disease, (3) chest wall disorder, (4) severe renal or liver dysfunction, (5) dementia and psychic disorder, (6) anemia, (7) severe renal or liver dysfunction, (8) abnormal shadows with a maximum diameter >50 mm on chest CT, and (9) other diseases potentially affecting respiratory function. Particularly, ILD and COPD were excluded from the study by imaging tests and LFTs.

## Lung function tests

All patients underwent LFTs, including spirometry, $D_{LCO}$ ($T_{LCO}$), $V_A$, residual volume, and total lung capacity, using a pulmonary function testing system (CHESTAC-8900®; CHEST Co., Ltd., Tokyo, Japan). Our hospital is sited 621 meters above sea level. The $D_{LCO}$, $K_{CO}$, and $V_A$ were measured by the single-breath method according to ERS and ATS standards to measure carbon monoxide gas transfer [11, 12]. The anatomical dead space was fixed at 150 ml to obtain reference values. We used $V_A$ reported in L (standard temperature and pressure, dry; STPD conditions) to obtain $D_{LCO}$.

In terms of diffusing capacity of the lungs for carbon monoxide notation, we have referred to the diffusivity of the traditional unit (ml / min / mmHg) as $D_{LCO}$ and of the SI unit system (mmol / min / kPa) as $T_{LCO}$.

## Pulmonary function test equations for $D_{LCO}$ measurements

We used the Nishida's and Burrows' equations for determining the percent predicted $D_{LCO}$($T_{LCO}$) and $K_{CO}$($D_{LCO}/V_A$), which are often used in daily clinical practice in Japan [13, 14]. We confirmed a normal percent predicted $D_{LCO}$ using previous linear equations because the predicted $D_{LCO}$ values obtained using each predictive linear model equation differ significantly [15].

## Statistical analysis

The values provided in the tables represent the mean ± standard deviation. All DLCO, VA, and KCO (DLCO/VA) data were converted to z-scores according to the GLI-2017 equations, assuming the GLI-2017 equations for Caucasians were applicable to Japanese. If the GLI-2017 equations provided a good fit, the derived DLCO z-scores were expected to be symmetric around zero [3, 16]. After generating GAMLSS modeling equations derived from the present data, we converted the data for the model assessment group to z-scores according to the present equations. Subsequently, we compared the current z-scores to those calculated by GLI-2017 using a one-sample t-test in the model assessment groups. The resulting z-scores had a mean of zero and a standard deviation of one, indicating that the data was reasonably well fitted if it was close to zero [16].

We developed separate prediction equations for the $D_{LCO}$ ($T_{LCO}$), $V_A$, and $K_{CO}$ ($D_{LCO}/V_A$), including age and height as potential predictors for men and women. We considered the following modeling strategies while developing the prediction equations: GAMLSS considers numerous residual distributions and provides several link functions between the predictors and outcomes, as well as the ability to integrate each moment's parameter predictors (including the median, variability, skewness, and kurtosis) [2, 17–20]. The GAMLSS includes the LMS method for establishing reference equations, which can be used to define the Box-Cox-Cole-Green (BCCG) residual distribution in the R package "GAMLSS." The BCCG is based on Cole and Green's pioneering work in fitting a single smoothing term to each of the three

distribution parameters [17]. The normal, BCCG, and Box-Cox-power-exponential (BCPE) distributions were all considered during the GAMLSS model development process [6, 9, 18]. We analyzed the log and identified link functions to determine the need for a predictor to model each moment parameter (median, the coefficient of variation, and skewness v), and its inclusion in the original or log style. To model the median μ (M moment of LMS), we considered the height and age as candidate predictors. We considered age as a candidate predictor while modeling variability (S moment of LMS) and skewness v (L moment of LMS).

The GAMLSS model with the lowest Bayesian information criterion (BIC) was selected as the best model. Given the importance of evaluating predictive performance, we chose 4/5 of the individuals to build the model and 1/5 to assess it. Computing the BIC values for the GLI-2017 prediction equations and previous linear prediction equations were practically impossible; therefore, we compared the performance of the "best" GAMLSS models and the GLI-2017 prediction equations for Japanese using mean squared errors (MSEs) [2, 17–20].

## Results

### Study population

The study cohort comprised 390 Japanese patients (193 men and 197 women) aged 16 to 85 years, with a maximum BMI <30 kg/m$^2$ (Table 1). Males involved in model assessment were

**Table 1. All parameters of the study population grouped according to their sex.**

| Parameters | | | | | | |
|---|---|---|---|---|---|---|
| | **Male** | | | **Female** | | |
| | **All** | **Individuals to build the model** | **Individuals for model assessment** | **All** | **Individuals to build the model** | **Individuals for model assessment** |
| Subjects, n | 193 | 154 | 39 | 197 | 160 | 37 |
| Age, year | 57.01 ±19.39 | 54.08±20.23 | 68.56±8.96 | 56.11 ±18.27 | 54.39±18.85 | 63.57±13.33 |
| Height, cm | 168.17 ±6.64 | 168.44±6.65 | 166.97±6.52 | 156.22 ±6.47 | 156.18±6.57 | 156.40±6.10 |
| Weight, kg | 65.35 ±9.48 | 65.25±9.79 | 65.72±8.24 | 54.32±8.30 | 54.36±8.33 | 54.12±8.28 |
| BMI, kg/m$^2$ | 23.10 ±2.99 | 22.99±3.10 | 21.07±1.95 | 22.27±3.22 | 22.30±3.23 | 22.13±3.22 |
| BSA, m$^2$ | 1.74±0.14 | 1.74±0.14 | 1.77±0.13 | 1.52±0.12 | 1.52±0.12 | 1.52±0.12 |
| Spirometry | | | | | | |
| VC, L | 3.99±0.80 | 4.06±0.82 | 3.74±0.65 | 2.92±0.57 | 2.93±0.60 | 2.84±0.46 |
| IC, L | 2.75±0.55 | 2.76±0.58 | 2.69±0.41 | 2.03±0.41 | 2.04±0.42 | 1.97±0.35 |
| FVC, L | 3.90±0.63 | 4.00±0.86 | 3.62±0.66 | 2.92±0.60 | 2.95±0.62 | 2.80±0.50 |
| FEV$_1$, L | 2.97±0.82 | 3.05±0.86 | 2.65±0.51 | 2.31±0.57 | 2.34±0.59 | 2.18±0.47 |
| | Male | | | Female | | |
| | All | Individuals to build the model | Individuals for model assessment | All | Individuals to build the model | Individuals for model assessment |
| FEV$_1$/FVC | 75.25 ±10.21 | 75.67±10.63 | 73.59±8.28 | 83.02±4.29 | 78.87±8.85 | 77.58±6.64 |
| PEF, L·s$^{-1}$ | 7.80±1.66 | 7.82±1.74 | 7.70±1.33 | 5.67±1.25 | 5.70±1.25 | 5.56±1.27 |
| Carbon monooxide transfer | | | | | | |
| T$_{LCO}$, mmol·min$^{-1}$·kPa$^{-1}$ | 8.51±1.68 | 8.71±1.75 | 7.68±1.06 | 7.04±1.16 | 7.08±1.16 | 6.89±1.15 |
| D$_{LCO}$, ml·min-$^1$·mmHg$^{-1}$ | 25.65 ±25.41 | 26.03±5.23 | 22.95±3.16 | 21.03±3.45 | 21.14±3.46 | 20.57±3.42 |

(*Continued*)

**Table 1.** (Continued)

| Parameters | | | | | | |
|---|---|---|---|---|---|---|
| | **Male** | | | **Female** | | |
| | **All** | **Individuals to build the model** | **Individuals for model assessment** | **All** | **Individuals to build the model** | **Individuals for model assessment** |
| IVC, L | 3.06±0.59 | 3.10±0.60 | 2.89±0.48 | 2.25±0.44 | 2.26±0.45 | 2.18±0.37 |
| RV, L | 1.71±0.44 | 2.15±0.54 | 1.90±0.43 | 1.34±0.29 | 1.72±0.39 | 1.38±0.23 |
| $V_A$, L | 4.66±0.75 | 4.69±0.78 | 4.57±0.61 | 3.60±0.64 | 3.63±0.66 | 3.47±0.53 |
| KCO (DLCO/$V_A$), ml·min$^{-1}$·mmHg$^{-1}$·L$^{-1}$ | 5.51±1.05 | 5.62±1.09 | 5.08±0.75 | 5.92±0.87 | 5.91±0.88 | 5.96±0.80 |
| | Male | | | Female | | |
| | All | Individuals to build the model | Individuals for model assessment | All | Individuals to build the model | Individuals for model assessment |
| Barometric pressure, mmHg | 716.12 ±22.01 | 715.33±21.49 | 718.99±23.84 | 713.04 ±19.33 | 713.97±20.83 | 709.04±10.01 |
| Breath holding time, s | 10.47 ±0.56 | 10.44±0.44 | 10.59±0.88 | 10.72±0.72 | 10.72±0.65 | 10.72±0.97 |
| $D_{LCO}$ predicted values (expressed as a percentage of predicted $D_{LCO}$ for various reference standards) | | | | | | |
| Nishida et al. [13] | 97.44 ±12.14 | 98.07±12.70 | 94.97±9.34 | 97.19 ±11.87 | 97.52±12.18 | 95.94±10.45 |
| Burrows et al. [14] | 129.67 ±22.70 | 128.61±23.80 | 134.29±17.00 | 121.44 ±17.68 | 129.20±33.40 | 125.10±20.72 |
| GLI-2017 | 101.16 ±12.57 | 101.62±13.17 | 99.35±9.78 | 111.67 ±13.52 | 111.63±12.68 | 111.83±16.89 |

Abbreviations: BMI, body mass index; BSA, body surface area; FVC, forced vital capacity; FEV$_1$, forced expiratory volume in 1 s; TLC, total lung capacity; VC, vital capacity; IC, inspiratory capacity; PEF, peak expiratory flow; D$_{LCO}$, diffusing capacity of the lungs for carbon monoxide; T$_{LCO}$: transfer factor for carbon monoxide; IVC: inspiratory vital capacity; RV: residual volume; V$_A$: alveolar volume; K$_{CO}$: carbon monoxide transfer coefficient; GLI-2017: Global Lung Function Initiative 2017 reference values [2]; Nishida et al.: Nishida et al. reference values [13]; and Burrows et al.: Burrows et al. reference values [14]. Data are presented as the mean ± standard deviation.

older and had lower vital capacity (VC), forced vital capacity (FVC), and forced expiratory volume in 1 s (FEV$_1$) levels than those involved in model building. Females involved in model assessment were older than those involved in model building, but there was no significant difference in each index of pulmonary function test. In order to evaluate the current study equations, we planned to randomly assign 1/5 of the patients to the model assessment group. Finally, 39 male individuals were evaluated as models (20.2% of the male population). Thirty-seven female individuals were involved in model evaluation (18.8% of the total female population). Table 2 summarizes the age distribution of model assessment participants, including the mean and 95% confidence intervals for D$_{LCO}$ and D$_{LCO}$/V$_A$ by age decade. The 95% confidence interval for D$_{LCO}$ was wider in the younger age group, as shown in Table 2, indicating more variation in D$_{LCO}$ than in the elderly. D$_{LCO}$/V$_A$ was higher in younger age groups and decreased in older age groups, whereas VA was less variable with age and increased with height. Conversely, it can be seen from Fig 2 that D$_{LCO}$/V$_A$ is inversely proportional to age, while D$_{LCO}$ and VA are height-dependent. The age of the participants (154 men and 160 women) used to develop equations for D$_{LCO}$ outcomes ranged as 16–85 years. The degree of D$_{LCO}$ varied according to the age and height, being greater in younger individuals than in older individuals. The D$_{LCO}$ (T$_{LCO}$) and V$_A$ were height-dependent.

To characterize our study population in relation to the GLI-2017, we calculated the z-scores for the D$_{LCO}$ (T$_{LCO}$), V$_A$, and DLCO/VA reference values as follows: the mean ± standard

**Table 2. Age distribution of study participants for the model assessment with mean and 95% confidence intervals for $D_{LCO}$ and $D_{LCO}/V_A$ stratified by their sex.**

| Age decade | Male (N = 154) | | | | Female (N = 160) | | | |
|---|---|---|---|---|---|---|---|---|
| | N (%) | $D_{LCO}$ (ml/min/mmHg) | $K_{CO}$ ($D_{LCO}/V_A$) (ml/min/mmHg/L) | $V_A$ (L) | N (%) | $D_{LCO}$ (ml/min/mmHg) | $D_{LCO}/V_A$ (ml/min/mmHg/L) | $V_A$ (L) |
| 15–19 years | 10 (6.5%) | 30.18 [27.74–32.63] | 6.62 [5.78–7.46] | 4.63 [4.25–5.00] | 3 (1.9%) | 23.81 [20.78–26.85] | 6.23 [5.08–7.37] | 3.91 [2.67–5.15] |
| 20–29 years | 14 (9.0%) | 30.93 [28.75–33.11] | 6.52 [5.75–7.30] | 4.87 [4.35–5.40] | 19 (11.9%) | 24.03 [21.98–26.09] | 6.63 [6.13–7.13] | 3.68 [3.32–4.04] |
| 30–39 years | 21 (13.6%) | 30.43 [28.44–32.43] | 6.07 [5.59–6.56] | 5.10 [4.66–5.55] | 19 (11.9%) | 23.00 [21.61–24.39] | 6.06 [5.82–6.30] | 3.82 [3.55–4.08] |
| 40–49 years | 18 (11.7%) | 29.39 [27.32–31.45] | 5.98 [5.58–6.37] | 4.96 [4.61–5.30] | 20 (12.5%) | 22.51 [21.45–23.57] | 6.02 [5.37–6.67] | 3.85 [3.51–4.19] |
| 50–59 years | 21 (13.6%) | 26.53 [24.54–28.52] | 5.56 [4.99–6.12] | 4.87 [4.51–5.23] | 27 (16.9%) | 22.13 [20.53–23.73] | 5.54 [5.20–5.88] | 4.05 [3.73–4.37] |
| 60–69 years | 24 (15.6%) | 23.79 [22.23–25.35] | 5.17 [4.80–5.54] | 4.67 [4.32–5.02] | 33 (20.6%) | 20.27 [19.46–21.08] | 5.89 [5.61–6.18] | 3.49 [3.29–3.70] |
| 70–79 years | 30 (19.5%) | 21.79 [20.65–22.93] | 5.09 [4.78–5.39] | 4.34 [4.07–4.61] | 24 (15.0%) | 18.65 [17.64–19.66] | 5.91 [5.59–6.22] | 3.19 [3.00–3.39] |
| 80–85 years | 16 (10.5%) | 20.25 [18.20–22.29] | 4.98 [4.44–5.52] | 4.13 [3.75–4.51] | 15 (9.3%) | 16.88 [15.79–17.97] | 5.36 [4.91–5.81] | 3.20 [2.91–3.49] |

Abbreviations: $D_{LCO}$: single-breath diffusing capacity for carbon monoxide; $D_{LCO}/V_A$: single-breath diffusing capacity for carbon monoxide per unit of lung volume; $K_{CO}$, carbon monoxide transfer coefficient; and $V_A$, alveolar volume.

error z-scores were -0.107 [-0.479, 0.482] ($D_{LCO}$), 1.000 [0.573, 2.327] ($K_{CO}$ [$D_{LCO}/V_A$]), and -1.617 [-2.214, -0.904] ($V_A$) for men and 0.586 [0.127, 1.154], 2.017 [1.472, 2.736], and -1.525 [-2.098, -0.852] for women (Fig 1). We tested the z-scores derived from the current and GLI-2017 equations using a one-sample t-test to examine the adequacy of applying the GLI-2017 equations to our data (Table 3). In the model assessment groups, almost all tests resulted in p-values $<10^{-2}$, except for the $D_{LCO}$ ($T_{LCO}$) for men, thus indicating a disagreement between our observed data and the GLI-2017 predicted values. This suggested that the GLI-2017 prediction equations were inappropriate for the Japanese population.

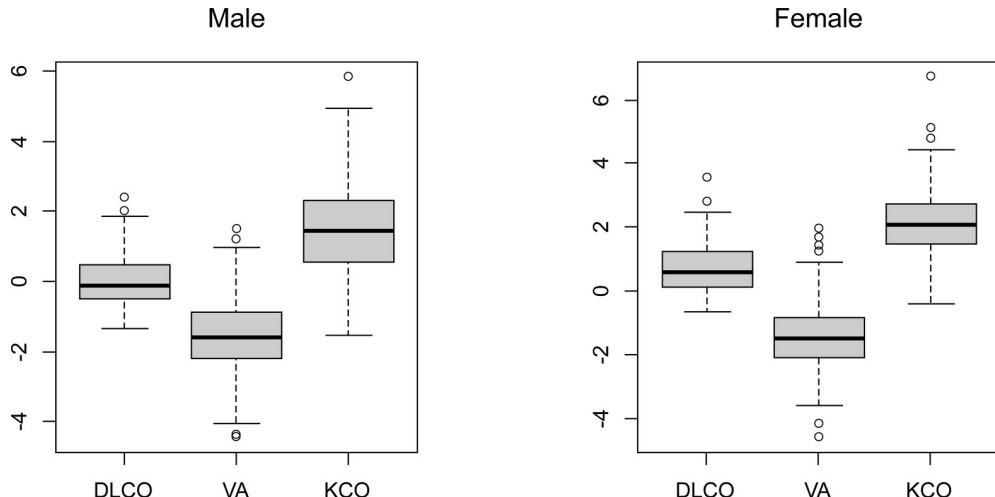

**Fig 1. Box plots of the $D_{LCO}$, $K_{CO}$ ($D_{LCO}/V_A$), and $V_A$ z-scores adjusted for the age and height, based on the Global Lung Function Initiative 2017 equations in 390 Japanese patients (193 men and 197 women).**
Abbreviations: $D_{LCO}$, single breath diffusing capacity for carbon monoxide; $K_{CO}$, single breath diffusing capacity for carbon monoxide per unit of lung volume; and $V_A$, alveolar volume.

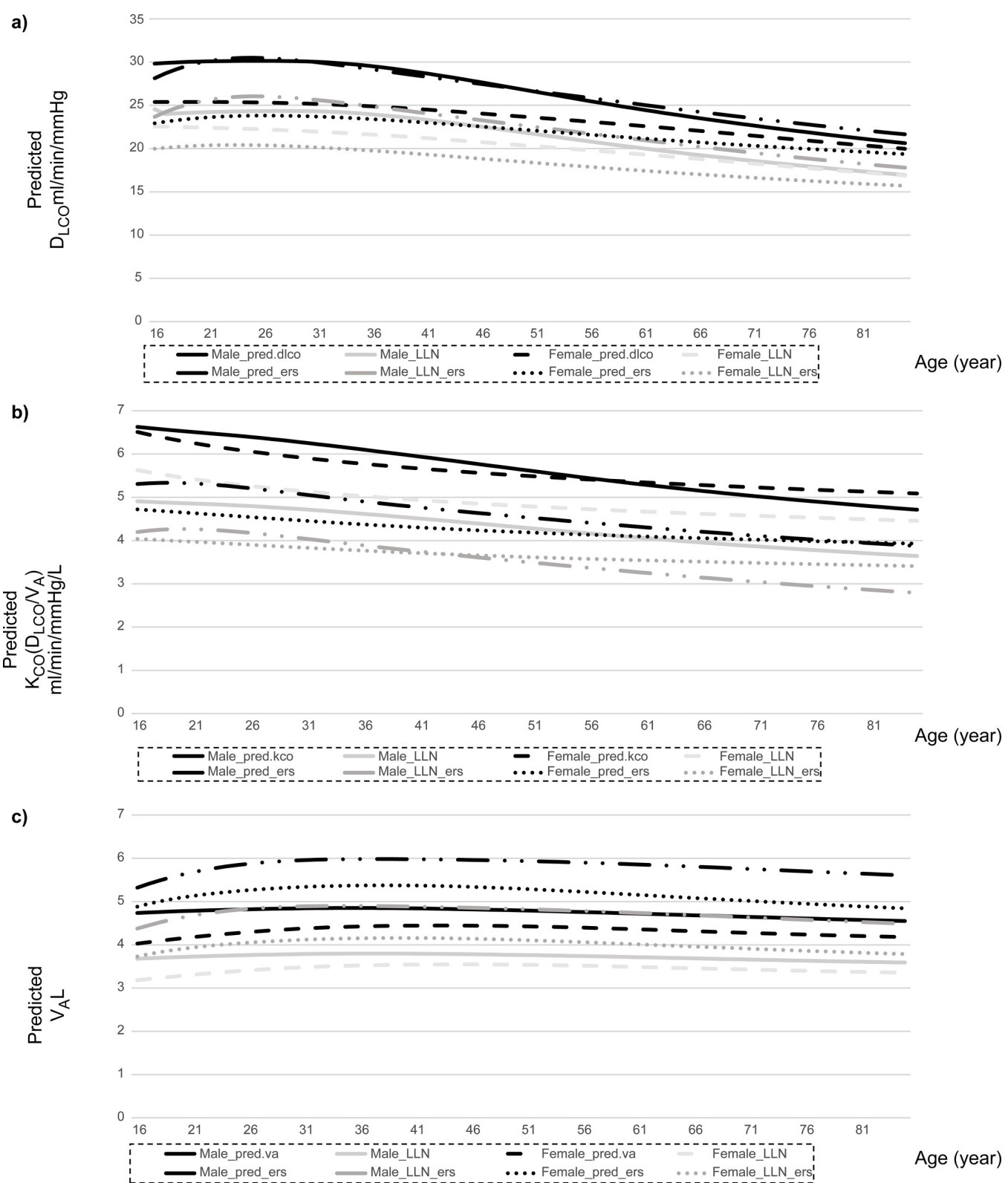

**Fig 2. Relationships between the present study values and GLI-2017 predicted values and the lower limits (LLN) of normal in Japanese population (n = 390), with a height of 170 cm and of different ages.** Abbreviations: $D_{LCO}$, single breath diffusing capacity for carbon monoxide; $K_{CO}$, single breath diffusing capacity for carbon monoxide per unit of lung volume; LLN, lower limits of normal; and $V_A$, alveolar volume.

## Reference values

Table 4 summarizes the best GAMLSS models of the $D_{LCO}$ ($T_{LCO}$) and $V_A$ in men and women, respectively. The height and age were independent predictors of each M ($\mu$), which required a natural logarithmic transformation of the height and a spline function for age, consistent with the GLI-2017 equations. We selected the BCCG distribution over the normal and BCPE distributions to model all prediction equations.

## Comparison with existing reference values

Fig 2 depicts the relationships between the present study and the GLI-2017 predicted values and lower limits of normal (LLN) in the Japanese population (n = 390) for a height of 170 cm and different ages. The GLI-2017 reference value for the $D_{LCO}$ was lower than our current values in women but was consistent with our values in men (Fig 2A). In both men and women, the GLI-2017 reference values for the $K_{CO}$ ($D_{LCO}/V_A$) were lower than our current values for all age decades (Fig 2B). However, the GLI-2017 reference value for $V_A$ was higher than our calculated value for all age decades (Fig 2C).

Fig 3 depicts the relationships between the present study and GLI-2017 predicted values and LLN in the Japanese population (n = 390) for participants aged 60 years and of different heights. In contrast to the $D_{LCO}$-age relationship, the GLI-2017 reference value for the $D_{LCO}$ in men was greater than our current values but was consistent with our current values in women, both aged >60 years with different heights (Fig 3A). In both men and women, the GLI-2017 reference value for the $K_{CO}$ ($D_{LCO}/V_A$) was lower than our current values across all heights (Fig 3B), whereas that for $V_A$ was greater across all heights (Fig 3C).

Fig 4 depicts the present study equation and previous reference equations for men and women with a height of 170 cm (60 kg, 1.69 $m^2$). Compared with previous linear $D_{LCO}$ reference values, our current values were within the range established by Nishida et al. (1976, Japan) and Burrows et al. (1961, USA) for all age decades. In other words, the current Japanese $D_{LCO}$ ($T_{LCO}$) values were certainly placed between the Japanese and Caucasian $D_{LCO}$ values (Fig 4). The current KCO ($D_{LCO}/V_A$) reference values were higher than those of previous

**Table 3. Z-scores for $D_{LCO}$, $K_{CO}$ and $V_A$ according to our current equation and the GLI-2017 equation for the Japanese population.**

| | Male (n = 39) | | | Female (n = 37) | | |
|---|---|---|---|---|---|---|
| Z-scores according to our current equation | | | | | | |
| | $zD_{LCO}$ | $zK_{CO}$ | $zV_A$ | $zD_{LCO}$ | $zK_{CO}$ | $zV_A$ |
| Mean | 0.006 | -0.141 | 0.061 | -0.005 | 0.305 | -0.269 |
| Standard Error | 0.110 | 0.136 | 0.124 | 0.196 | 0.154 | 0.15 |
| min | -1.141 | -1.865 | -1.718 | -2.129 | -1.544 | -2.462 |
| max | 1.698 | 1.892 | 1.495 | 3.116 | 2.375 | 2.402 |
| Z-scores according to the GLI-2017 equation | | | | | | |
| | $zD_{LCO}$ | $zK_{CO}$ | $zV_A$ | $zD_{LCO}$ | $zK_{CO}$ | $zV_A$ |
| Mean | 0.017 | 1.27 | -1.439 | 0.599 | 2.229 | -1.537 |
| Standard Error | 0.110 | 0.16 | 0.136 | 0.114 | 0.168 | 0.162 |
| min | -0.793 | -0.557 | -3.246 | -0.660 | 0.415 | -4.154 |
| max | 2.423 | 3.38 | 0.313 | 1.933 | 4.78 | 1.661 |
| p-value | 0.94390 | 0.00000 | 0.00000 | 0.00966 | 0.00000 | 0.00000 |

Abbreviations: GLI, global lung function initiative; $D_{LCO}$, single breath diffusing capacity for carbon monoxide; $D_{LCO}/V_A$, single breath diffusing capacity for carbon monoxide per unit of lung volume; $K_{CO}$, carbon monoxide transfer coefficient; and $V_A$, alveolar volume. Z-scores were significantly different between participants using our current prediction equations and the GLI-2017 equation (p<0.05; one sample t-test was used).

**Table 4. Corrected equations for the predicted values for the median (M), the variability around the median (S), and the skewness (L) for each of the $D_{LCO}$ test outcomes.**

| | M | S | L |
|---|---|---|---|
| Male | | | |
| $T_{LCO}$ mmol/min/kPa | exp[-4.15119+1.42677·ln[height]-0.26486·ln[age]+Mspline] | exp[-3.1864+0.2488·ln[age]+Sspline] | -2.4 |
| $D_{LCO}$ ml/min/mmHg | exp[-3.05697+1.42677·ln[height]-0.26486·ln[age]+Mspline] | exp[-3.1864+0.2488·ln[age]+Sspline] | -2.4 |
| $K_{CO}$ ($D_{LCO}$/$V_A$) ml/min/mmHg/L | exp[5.35455-0.53705·ln[height]-0.22823·ln[age]+Mspline] | exp[-1.47917-0.09025·ln[age]+Sspline] | 0.2 |
| $V_A$ L | exp[-8.44716+1.9724·ln[height]-0.03319·ln[age]+Mspline] | exp[-1.81187-0.03616·ln[age]+Sspline] | 0.3 |
| | M | S | L |
| Female | | | |
| $T_{LCO}$ mmol/min/kPa | exp[-3.80015+1.26398·ln[height]-0.16611·ln[age]+Mspline] | exp[-2.02125-0.07081·ln[age]+Sspline] | -2.3 |
| $D_{LCO}$ ml/min/mmHg | exp[-2.70593+1.26398·ln[height]-0.16611·ln[age]+Mspline] | exp[-2.02125-0.07081·ln[age]+Sspline] | -2.3 |
| $K_{CO}$ ($D_{LCO}$/$V_A$) ml/min/mmHg/L | exp[5.78602-0.68239·ln[height]-0.14728·ln[age]+Mspline] | exp[-2.2351+0.05624·ln[age]+Sspline] | -0.5 |
| $V_A$ L | exp[-10.338797+2.300958·ln[height]-0.003541·ln[age]+Mspline] | exp[-1.74859-0.04783·ln[age]+Sspline] | -0.5 |

Mspline and Sspline correspond to the age-variable coefficients from the look-up tables provided in the S1 Data. The height and age are expressed as cm and years, respectively.

Predicted value: M; the lower limit of normal [5th percentile]: exp[ln[M]+ln[1−1.645·L·S]/L]; the upper limit of normal [5th percentile]: exp[ln[M]+ln[1+1.645·L·S]/L]; z-score: [[measured/M]L-1]/[L·S]; %predicted: [measured/M]·100; exp: natural exponential; and ln: natural logarithm.

equations, whereas the current $V_A$ reference values were lower than GLI-2017 in both men and women (Fig 4). The S1 File illustrates the differences between our current mean reference values and those from the GLI-2017, Nishida et al., and Burrows et al. (S1 Fig in S1 File).

Subsequently, we compared the predictive performance between our study and previous equations in terms of MSEs (Table 5). We observed smaller MSEs for the $D_{LCO}$, $K_{CO}$ ($D_{LCO}$/$V_A$), and $V_A$ than those from the GLI-2017 equation, which suggested better predictive results from our current equations to the Japanese population set.

## Discussion

This is the first study to model the $D_{LCO}$ using the GAMLSS approach in patients with near-normal lung function and to assess the applicability of the GLI-2017 prediction equation for a Japanese patient cohort. First, the GLI-2017 prediction equation for Caucasian patients did not match the $D_{LCO}$, $K_{CO}$ ($D_{LCO}$/$V_A$), and $V_A$ data for the contemporary Japanese patient population, thereby highlighting the importance of developing prediction equations for the Japanese population. Second, we established prediction equations for the $D_{LCO}$, $K_{CO}$ ($D_{LCO}$/$V_A$), and $V_A$ in a Japanese population aged 16–85 years using the GAMLSS model. Third, the GAMLSS model outperformed GLI-2017 equations and previous linear regression equations for the Japanese population.

Upon applying the GLI-2017 prediction equation for Caucasians to our study cohort (Fig 1), the z-scores of the $D_{LCO}$ ($T_{LCO}$) were relatively nearer zero, particularly in men. However, the z-scores of the $K_{CO}$ ($D_{LCO}$/$V_A$) were significantly higher (1.51 ± 0.090) in men and women, whereas those for $V_A$ were significantly lower (-1.566 ± 0.075). Thus, the GLI-2017

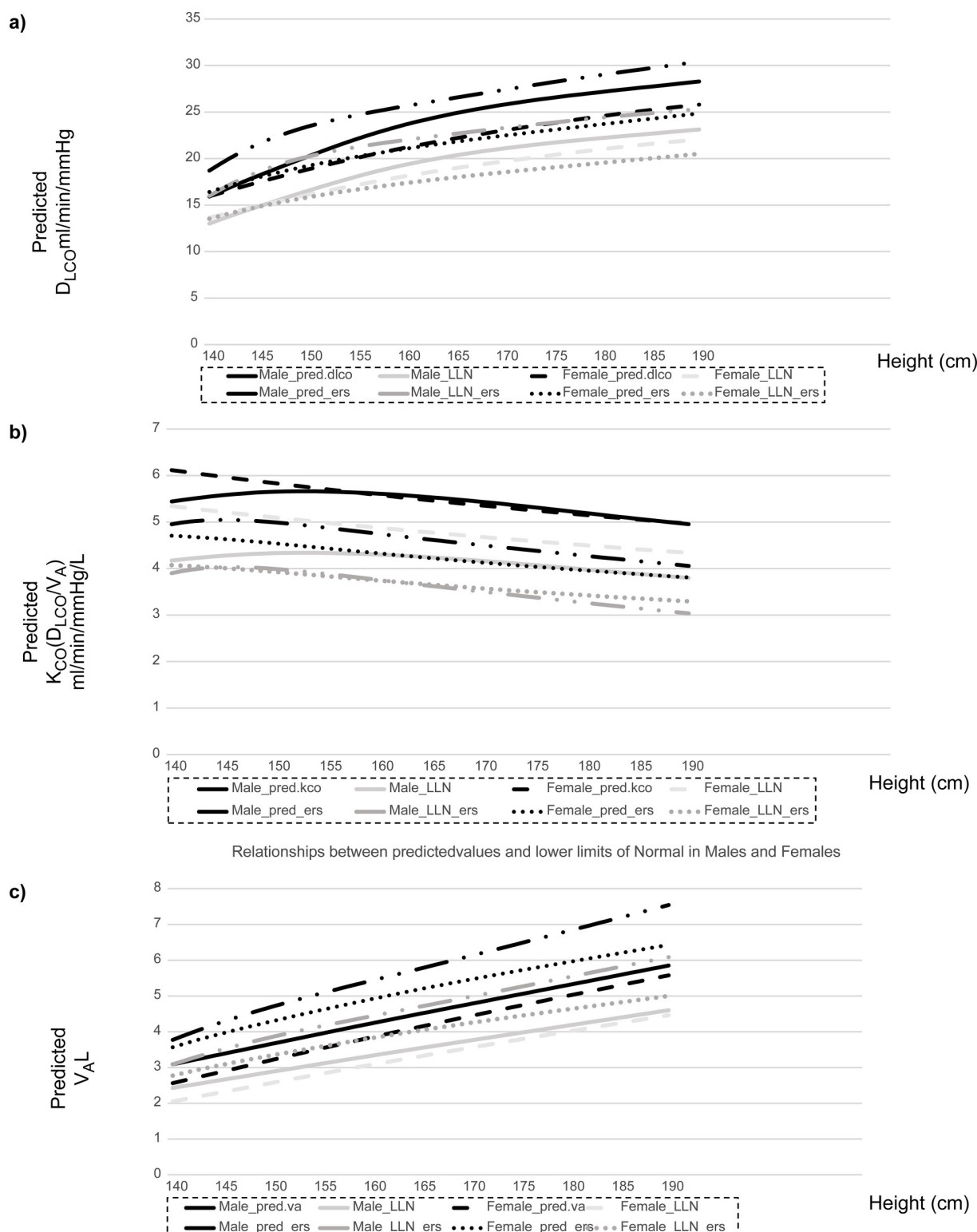

**Fig 3. Relationship between the current values and GLI-2017 predicted values and the lower limits of normal in Japanese population (n = 390), aged 60 years and of different heights.** Abbreviations: $D_{LCO}$, single breath diffusing capacity for carbon monoxide; $K_{CO}$, single breath diffusing capacity for carbon monoxide per unit of lung volume; LLN, lower limits of normal; and $V_A$, alveolar volume.

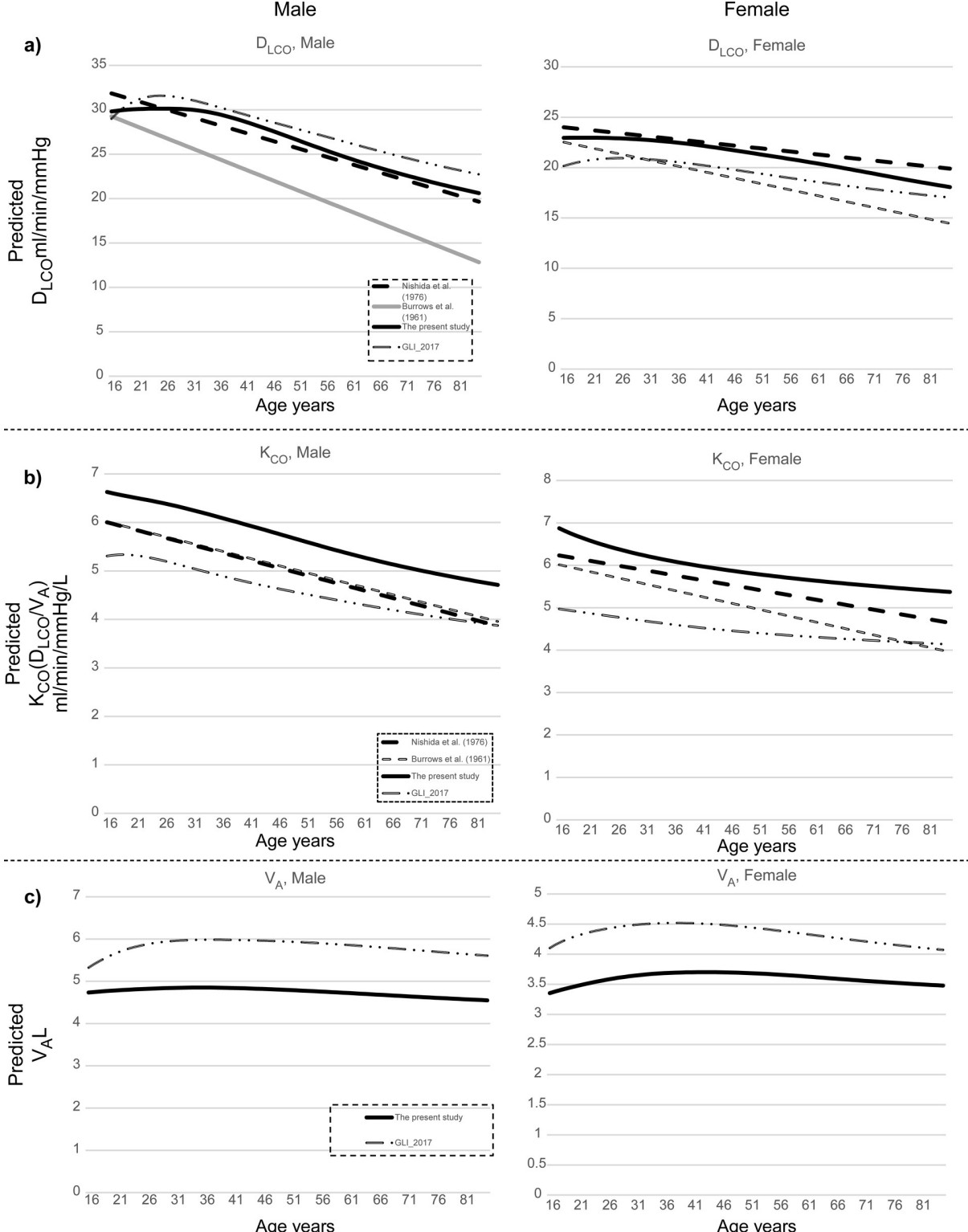

**Fig 4. Comparison of reference equations in men.** Predicted a) single breath diffusing capacity for carbon monoxide ($D_{LCO}$), b) single breath diffusing capacity for carbon monoxide per unit of lung volume ($D_{LCO}/V_A$), and c) alveolar volume ($V_A$). Abbreviations: $D_{LCO}$, single breath diffusing capacity for carbon monoxide; $K_{CO}$, single breath diffusing capacity for carbon monoxide per unit of lung volume; and $V_A$, alveolar volume.

**Table 5. Mean square errors of the $D_{LCO}$, $K_{CO}$, $V_A$ from our current equations, the GLI-2017 equation, and those by Nishida et al. and Burrows et al.**

| | | Male (n = 39) | | | Female (n = 37) | | |
|---|---|---|---|---|---|---|---|
| | | $D_{LCO}$ | $K_{CO}$ ($D_{LCO}/V_A$) | $V_A$ | $D_{LCO}$ | $K_{CO}$ ($D_{LCO}/V_A$) | $V_A$ |
| Current study | GAMLSS | 2.132 | 0.674 | 0.497 | 3.097 | 0.824 | 0.474 |
| GLI-2017 | GAMLSS | 2.255 | 1.145 | 1.117 | 3.711 | 1.819 | 0.958 |
| Nishida et al. | Linear model | 2.290 | 0.971 | - | 3.098 | 1.128 | - |
| Burrows et al. | Linear model | 6.082 | 0.920 | - | 5.015 | 1.572 | - |

Abbreviations: GLI, global lung function initiative; DLCO, single breath diffusing capacity for carbon monoxide; DLCO/VA, single breath diffusing capacity for carbon monoxide per unit of lung volume; KCO, carbon monoxide transfer coefficient; VA, alveolar volume; GLI-2017: Global Lung Function Initiative 2017 reference values [2]; Nishida: Nishida et al. reference values [13]; and Burrows: Burrows et al. reference values [14].

prediction equation tended to underestimate the $K_{CO}$ ($D_{LCO}/V_A$) and overestimate the $V_A$, thus resulting in a relatively accurate estimate of DLCO in the Japanese population (Figs 2 and 3). Our results could be explained by the degree of ethnic heterogeneity in the GLI-2017 prediction equations for the Japanese population. These equations were derived from several Caucasian ethnicities [2, 3]. This observation holds true for the DLCO and the KCO and VA, which must be determined to ensure correct prediction.

The majority of studies in Asian populations and parts of Caucasian populations, including Japan, have used linear regression models to generate prediction equations for the $D_{LCO}$, despite the linearity assumption not always holding true for the relationships between the age, height, and the $D_{LCO}$ [21–23]. Our findings added to the growing body of evidence that the $D_{LCO}$ indices decrease nonlinearly with increasing age in ranging from 16 years to 85 years. The GAMLSS method addressed the previously mentioned issue by improving its ability to account for non-linear relationships between the age, height, and $D_{LCO}$ indices. Our results indicated that accounting for age and height as potential predictors in $D_{LCO}$ models was consistent with previously reported GAMLSS prediction models in the Caucasian population. For example, Verbanck et al. demonstrated that the $T_{LCO}$ ($D_{LCO}$) decreases monotonically across the entire age range of a healthy Caucasian population, with the variability remaining nearly constant between 20 and 80 years, close to the age of our study population [24].

According to the ATS and ERS, several factors affect pulmonary function, including age, sex, height, weight, and ethnic origin [2, 7]. In our Japanese population, we developed prediction equations for the $D_{LCO}$ indices and the corresponding LLN. We presented a calculator in which the users could enter their age and desired height and immediately obtain the corresponding predictive values, z-scores, and LLN for the $D_{LCO}$ ($T_{LCO}$), $K_{CO}$ ($D_{LCO}/V_A$), and $V_A$, respectively (S1 Data).

This study had several strengths. First, we established reference values for the $D_{LCO}$ through an analysis using the GAMLSS method on Japanese participants aged 16–85 years. Second, the secular trends in pulmonary function characteristics warrant periodically updated reference values for normal and abnormal classifications to reflect contemporary population realities. Our prediction equations were based on a representative sample of contemporary Japanese patients with near-normal lung function. The GAMLSS reference values for $D_{LCO}$ in the Japanese population were unavailable prior to the current study.

However, our study had some limitations. We compared our reference values to the GLI-2017 values for the Caucasian population owing to the absence of a GAMLSS equation for the $D_{LCO}$ in the Japanese population. Second, this retrospective observational design made it impossible to identify all patients with normal CT screening results. Moreover, we could not recruit healthy volunteers because the collection of raw LFT data is strictly regulated for

Japanese patients. Third, we collected data from a single laboratory, whereas the sample size used to build the model was comparable to that used in previous reports [18, 21]. There is a possibility that factors such as the small sample size, single-center measurements, the altitude of our hospital, and method of determining anatomical dead space contributed to discrepancies between our predictions and those of other researchers. As mentioned in the method, to ensure a sufficient sample size, patients with early-stage lung cancer, sarcoidosis, or asthma, with small abnormal shadows that did not meet the exclusion criteria, or without abnormal shadows were included in the CT screening. The fact that lung function data of these patients were used to create and validate the predictive formula may have caused our predictive formula to differ from other predictive formulas.

In conclusion, our current reference values based on the Japanese population were more appropriate for our sample than the GLI-2017 values, and differences between the two equations are attributed to the underestimation of the $K_{CO}$ ($D_{LCO}/V_A$) and the overestimation of $V_A$. Our study examined the effect of future application of new reference values based on the GAMLSS model equations for the assessment of $D_{LCO}$ in the Japanese population.

## Supporting information

**S1 File. Instruction for using the DLCO excel sheet and examples of calculating the predictive values.**
(DOCX)

**S1 Data. The DLCO excel sheet for calculation.**
(XLSX)

## Acknowledgments

We would like to thank Editage (www.editage.com) for English language editing.

## Author Contributions

**Data curation:** Yosuke Wada, Norihiko Goto, Yoshiaki Kitaguchi, Masanori Yasuo, Masayuki Hanaoka.

**Formal analysis:** Yosuke Wada.

**Investigation:** Yosuke Wada.

**Methodology:** Yosuke Wada.

**Project administration:** Masayuki Hanaoka.

**Writing – original draft:** Yosuke Wada.

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
