## [Decision Letter · Decision Letter 0]

10 May 2022

PONE-D-22-00545Referential equations for pulmonary diffusing capacity using GAMLSS models derived from Japanese individuals with near-normal lung functionPLOS ONE

Dear Dr. Wada,

Thank you for submitting your manuscript to PLOS ONE. After careful consideration, we feel that it has merit but does not fully meet PLOS ONE’s publication criteria as it currently stands. Therefore, we invite you to submit a revised version of the manuscript that addresses the points raised during the review process.

We look forward to receiving your revised manuscript.

Kind regards,

Aleksandra Barac

Academic Editor

PLOS ONE

Journal Requirements:

Reviewers' comments:

Reviewer's Responses to Questions

**Comments to the Author**

1. Is the manuscript technically sound, and do the data support the conclusions?

Reviewer #1: Yes

Reviewer #2: Yes

Reviewer #3: Yes

2. Has the statistical analysis been performed appropriately and rigorously? 

Reviewer #1: I Don't Know

Reviewer #2: Yes

Reviewer #3: Yes

3. Have the authors made all data underlying the findings in their manuscript fully available?

Reviewer #1: Yes

Reviewer #2: Yes

Reviewer #3: Yes

4. Is the manuscript presented in an intelligible fashion and written in standard English?

Reviewer #1: Yes

Reviewer #2: Yes

Reviewer #3: Yes

5. Review Comments to the Author

Reviewer #1: it is an interesting study, insofar as it could contribute to having reference equations for the Japanese and even Asian population.

the result of your study indicates that using the reference equations of GLI-2017, the KCO has been overestimated and the VA has been underestimated. the main difference is therefore the overestimation of the VA. and this is mainly due to methodological imperfections such as :

- This is a single center study

- the sample is small

- line (89-91 and line 95) : you included hypertensive and other with early stage lung cancer, sarcoidosis, or asthma and small abnormal shadows

- TLCO data were not adjusted to the inspiratory oxygen partial pressure at standard barometric pressure

- the altitude of the centre in which the reference values were obtained was not mentioned

- you did not specify whether the reference values were obtained using a fixed dead space correction of 150 mL or not.

Reviewer #2: The aim of this study was to establish appropriate reference equations of diffusing capacity of the lung for carbon monoxide (DLCO), alveolar volume (VA), and transfer coefficient of the lung for carbon monoxide (KCO) in Japanese population. By using the GAMLSS model on the data of pulmonary diffusion capacity tests collected from age 16-85 Japanese people, the authors proposed equations for calculating predictive values of DLCO, VA and KCO in Japanese without chronic lung disease. The authors also compared the predictive values derived from their equations with those values from other reference equations to examine the performance of their prediction equations.

There are some questions and concerns need to be clarified:

1.Line 84. Materials and Methods. It would be better to provide objective parameters for some of the including and excluding criteria, e.g. anemia, severe renal or liver dysfunction, etc. Besides, “not anemia” of inclusion criteria could be omitted since “anemia” had been listed as one of the exclusion criteria.

2.Line 86. Materials and Methods. The authors need to clarify the meaning of inclusion criteria (6): “ …chest computed tomography (CT) performed on the day of LFTs in the second half of the previous year”. Did it mean that “chest CT performed within 6 moths before the lung function test”?

3.Line 159 and Table 1. The number of women randomized for model assessment (37) was not equal to 1/5 of total recruited woman participant, as the authors mentioned in the text.

4.Line 155. Results. Was there any significantly statistic finding among the data from different groups in Table 1?

5.Line 161. Results. The authors need to clarify how did they determine that DLCO variation was greater in younger individuals, and DLCO and VA were height-dependent, based on the data of Table 2.

6.The intents in Figure 2, 3, and 4 were too blurry to be read and interpreted. The authors need to revise these figures for readers’ convenience and reviewing.

7.The authors used “DLCO (TLCO)”, “DLCO” and “TLCO” in the text to express diffusing capacity for carbon monoxide (transfer factor for carbon monoxide), all of which indicated the same test. It was a little bit confused that these two terms (DLCO and TLCO) had different M equation in Table 4. Was this only for different expression of measurement unit (mmol/min/kPa vs. ml/min/mmHg), or for any other specific purpose?

Reviewer #3: The authors have investigated reference values in DLCO, VA and KCO in a Japanese reference population. The manuscript is informative and the statistic procedures well documented. I would ask for these revisions:

In the abstract, the following phrase is confounding, please change it if you want to: “were attributed to underestimation and overestimation of KCO (DLCO/VA) and VA, respectively, by the GLI-2017 for the Japanese population.”

It might be beneficial to add something like “transfer coefficient of the lungs for carbon monoxide (KCO)”, pulmonary function testing, and so forth to the keywords.

The introduction including explanation of methods was very clear and concise.

In Methods, you describe that patients can withdraw their data, however, would that still be possible if data is de-identified/anonymous? Please change or delete this information.

You mentioned several diseases that led to exclusion of the participant. To exclude bias, it would be highly interesting to know what kind of diseases your reference group have if they were not healthy volunteers nor have the diseases cited in the list. However, I can see the restrictions of a de-identified retrospective cohort, if this assessment is not feasible.

The figures are illustrative and clear. The second figure seems fuzzy, please upload a higher-pixel version if possible.

The manuscript is of interest and it is well written, highlighting the guideline-mentioned ethnic differences in lung function reference values.

6. PLOS authors have the option to publish the peer review history of their article (what does this mean?). If published, this will include your full peer review and any attached files.

Reviewer #1: **Yes: **KHADIJA AYED

Reviewer #2: No

Reviewer #3: No

---

## [Author Response · Author response to Decision Letter 0]

17 Jun 2022

We have revised the manuscript in accordance with reviewers’ comments in a point-by-point manner.

Reviewer #1

C. the result of your study indicates that using the reference equations of GLI-2017, the KCO has been overestimated and the VA has been underestimated. the main difference is therefore the overestimation of the VA. and this is mainly due to methodological imperfections such as :

1.- This is a single center study

2.- the sample is small

3.- line (89-91 and line 95) : you included hypertensive and other with early stage lung cancer, sarcoidosis, or asthma and small abnormal shadows

4.- TLCO data were not adjusted to the inspiratory oxygen partial pressure at standard barometric pressure

5.- the altitude of the centre in which the reference values were obtained was not mentioned

6.- you did not specify whether the reference values were obtained using a fixed dead space correction of 150 mL or not.

1.- This is a single center study

2.- the sample is small

3.- line (89-91 and line 95) : you included hypertensive and other with early stage lung cancer, sarcoidosis, or asthma and small abnormal shadows

Response

Thank you for your insightful comment. We agree with you and have added the following sentences to the Discussion section. Pages 34-35, Lines 359－368.

There is a possibility that factors such as the small sample size, single-center measurements, the altitude of our hospital, and method of determining anatomical dead space contributed to discrepancies between our predictions and those of other researchers. As mentioned in the method, to ensure a sufficient sample size, patients with early-stage lung cancer, sarcoidosis, or asthma, with small abnormal shadows that did not meet the exclusion criteria, or without abnormal shadows were included in the CT screening. The fact that lung function data of these patients were used to create and validate the predictive formula may have caused our predictive formula to differ from other predictive formulas.

4.- TLCO data were not adjusted to the inspiratory oxygen partial pressure at standard barometric pressure

Response: Thank you for your observation. We agree with you and have added the following sentence to the Methods section. Page 8, Lines 112－113

We used VA reported in L (standard temperature and pressure, dry; STPD conditions) to obtain DLCO.

5.- the altitude of the centre in which the reference values were obtained was not mentioned

Response: Thank you for your comment. We agree with you and have added the following sentence to Page 7, Lines 108－109, Lung function tests.

Our hospital is sited 621 meters above sea level.

6.- you did not specify whether the reference values were obtained using a fixed dead space correction of 150 mL or not.

Response: Thank you for pointing this out. We have added the following sentence to Page 8, Lines 111－112, Lung function tests.

The anatomical dead space was fixed at 150 ml to obtain reference values.

Reviewer #2

1.Line 84. Materials and Methods. It would be better to provide objective parameters for some of the including and excluding criteria, e.g. anemia, severe renal or liver dysfunction, etc. Besides, “not anemia” of inclusion criteria could be omitted since “anemia” had been listed as one of the exclusion criteria.

Response: Thank you for your good advice. We agree with you. We add the following sentences in Page 7, Line 101, Materials and Methods, Study participants.

(7) severe renal or liver dysfunction. 

Furthermore, in Page 6, Line 86, we have excluded "not anemia" from the inclusion criteria. 

2. Line 86. Materials and Methods. The authors need to clarify the meaning of inclusion criteria (6): “ …chest computed tomography (CT) performed on the day of LFTs in the second half of the previous year”. Did it mean that “chest CT performed within 6 months before the lung function test”?

Response: Thank you for your observation. We have revised the inclusion criteria into the following text: (5) no abnormality or localized shadow based on chest computed tomography (CT) performed within 6 months before the lung function test. Lines 91-92

3. Line 159 and Table 1. The number of women randomized for model assessment (37) was not equal to 1/5 of total recruited woman participant, as the authors mentioned in the text.

Response: Thank you for your good advice. We have revised the inclusion criteria into the following text: In order to evaluate the current study equations, we planned to randomly assign 1/5 of the patients to the model assessment group. Finally, 39 male individuals were evaluated as models (20.2% of the male population). Thirty-seven female individuals were involved in model evaluation (18.8% of the total female population). Lines 171-175

4. Line 155. Results. Was there any significantly statistic finding among the data from different groups in Table 1?

Response: Thank you for your observation. We have added the following paragraph to highlight the statistical significance of the findings: Males involved in model assessment were older and had lower vital capacity (VC), forced vital capacity (FVC), and forced expiratory volume in 1 s (FEV1) levels than those involved in model building. Females involved in model assessment were older than those involved in model building, but there was no significant difference in each index of pulmonary function test. Lines 166-170

5. Line 161. Results. The authors need to clarify how did they determine that DLCO variation was greater in younger individuals, and DLCO and VA were height-dependent, based on the data of Table 2.

Response: Thank you for your insightful comment. We have revised the inclusion criteria into the following text:

The 95% confidence interval for DLCO was wider in the younger age group, as shown in Table 2, indicating more variation in DLCO than in the elderly. DLCO/VA was higher in younger age groups and decreased in older age groups, whereas VA was less variable with age and increased with height. Conversely, it can be seen from Fig 2 that DLCO/VA is inversely proportional to age, while DLCO and VA are height-dependent. Line 177-183

6. The intents in Figure 2, 3, and 4 were too blurry to be read and interpreted. The authors need to revise these figures for readers’ convenience and reviewing.

Response: Thank you for your good advice. We have revised Figures 2, 3, and 4 to make them more readable.

7. The authors used “DLCO (TLCO)”, “DLCO” and “TLCO” in the text to express diffusing capacity for carbon monoxide (transfer factor for carbon monoxide), all of which indicated the same test. It was a little bit confused that these two terms (DLCO and TLCO) had different M equation in Table 4. Was this only for different expression of measurement unit (mmol/min/kPa vs. ml/min/mmHg), or for any other specific purpose?

Response: Thank you for your comment. As you noted, this is because the DLCO notation includes both SI units and traditional notations. When the diffusing capacity is expressed using the SI unit system, it is written as TLCO (mmol / min / kPa), whereas it is written as DLCO (ml / min / mmHg) using the traditional unit system.

 In Lines 107-109, we added the following paragraph: In terms of diffusing capacity of the lungs for carbon monoxide notation, we have referred to the diffusivity of the traditional unit (ml / min / mmHg) as DLCO and of the SI unit system (mmol / min / kPa) as TLCO.

Reviewer 3

1. In the abstract, the following phrase is confounding, please change it if you want to: “were attributed to underestimation and overestimation of KCO (DLCO/VA) and VA, respectively, by the GLI-2017 for the Japanese population.”

It might be beneficial to add something like “transfer coefficient of the lungs for carbon monoxide (KCO)”, pulmonary function testing, and so forth to the keywords.

Response: Thank you for pointing this out. We have revised the abstract as follows : Reference values obtained in this study were more appropriate for our sample than those reported in GLI-2017. Differences between the two equations were attributed to underestimating KCO (DLCO/VA) and overestimating VA, respectively, by the GLI-2017 for the Japanese population.

Lines 35-38

And we added Keywords as follows: “transfer coefficient of the lungs for carbon monoxide (KCO)”, “pulmonary function test.”

2. The introduction including explanation of methods was very clear and concise.

Response: Thank you for your kind comments.

3. In Methods, you describe that patients can withdraw their data, however, would that still be possible if data is de-identified/anonymous? Please change or delete this information.

Response: Thank you for your good advice. We agree with you. 

4. You mentioned several diseases that led to exclusion of the participant. To exclude bias, it would be highly interesting to know what kind of diseases your reference group have if they were not healthy volunteers nor have the diseases cited in the list. However, I can see the restrictions of a de-identified retrospective cohort, if this assessment is not feasible.

Response: Thank you for your kind comments. 

5. The figures are illustrative and clear. The second figure seems fuzzy, please upload a higher-pixel version if possible.

Response: Thank you for your kind comments.

We have formatted the figures in our manuscript according to the requirements of the journal.

---

## [Editor Report · Decision Letter 1]

24 Jun 2022

Referential equations for pulmonary diffusing capacity using GAMLSS models derived from Japanese individuals with near-normal lung function

PONE-D-22-00545R1

Dear Dr. Wada,

We’re pleased to inform you that your manuscript has been judged scientifically suitable for publication and will be formally accepted for publication once it meets all outstanding technical requirements.

Kind regards,

Aleksandra Barac

Academic Editor

PLOS ONE

---

## [Editor Report · Acceptance letter]

12 Jul 2022

PONE-D-22-00545R1 

Referential equations for pulmonary diffusing capacity using GAMLSS models derived from Japanese individuals with near-normal lung function 

Dear Dr. Wada:

I'm pleased to inform you that your manuscript has been deemed suitable for publication in PLOS ONE. Congratulations! Your manuscript is now with our production department. 

Kind regards, 

on behalf of

Dr. Aleksandra Barac 

Academic Editor

PLOS ONE